# Influence of Enterprise's Factor Inputs and Co-Opetition Relationships to Its Innovation Output

Lei Shi [1], Shan Gao [2,*], Airong Xu [2], Kexin Zheng [2], Yuanpeng Ji [3], Xianlei Dong [2] and Lizhi Xing [3,*]

[1] National Academy of Innovation Strategy, China Association for Science and Technology, Beijing 100038, China
[2] School of Business, Shandong Normal University, Jinan 250358, China
[3] College of Economics and Management, Beijing University of Technology, Beijing 100124, China
* Correspondence: 2020020979@stu.sdnu.edu.cn (S.G.); koken@bjut.edu.cn (L.X.)

**Abstract:** In the context of economic globalization, innovation has become a major drive for the sustainable development of enterprises, which emphasizes the importance of studying the influencing factors of enterprise innovation output. The purpose of this study is to clarify the influence mechanism of different indicators on enterprise innovation output, and then provide relevant suggestions for improving enterprise innovation ability. This paper takes 562 enterprises in Chaoyang Sub-park and Fengtai Sub-park of Zhongguancun in Beijing within the time span between 2015 and 2016 as the research objects, and constructs a comprehensive indicator system of influencing factors for enterprise innovation output from the perspective of enterprise co-opetition relationship, factor input and environmental factors. A quantitative model of innovation output and influencing factors was built and then solved, via spike-and-slab sparse function and stepwise regression, aiming at analyzing the influence of different indicators on enterprise innovation output. In addition, this paper also classifies enterprises according to their innovation level and explores the influence of indicators on different types of enterprises. The innovation of this study lies in the modeling of competition and cooperation between enterprises and the establishment of a relatively comprehensive indicator system of influencing factors for enterprise innovation output. The results show that the degree of technological collaboration between enterprises, the level of financing and the degree of capital and labor input in innovation activities have significant positive effects on enterprise innovation output. On the contrary, product and service competition, as well as capital competition, tends to have a negative impact, which urges enterprises to pay heed to the intensity of competition faced by products and services as well as the diversity of financing sources and investment targets to reduce the negative impacts. In addition, enterprises with different levels of innovation should take customized measures in terms of factor input and co-opetition relationships, in that some indicators such as network structure indicators negatively influence the innovation output of enterprises with lower levels of innovation, but has positive impacts on those with higher levels of innovation.

**Keywords:** innovation output; co-opetition; factor input; market environment; multi-influencing factor model

## 1. Introduction

In the age of knowledge economy, competition among enterprises is becoming increasingly intense. Against the backdrop of international factor movements, innovation has become a key force for enterprise growth, meaning that the innovation output of an enterprise speaks for its innovation capability. This paper examines the main influencing factors of enterprises' innovation output and discusses how these factors affect their innovation performance.

The Global Innovation Index (GII) 2020 report shows that China leapfrogged from the 29th place in 2015 to the 14th in 2020, being the only middle-income economy in the

GII top 30 [1]. Currently, China's economy is transformed from an investment-driven mode to an efficiency- and innovation-driven one, showing a new trend of high-quality development featuring "accelerated speed, optimized structure and shifted growth engine". In the Fifth Plenary Session of the 19th Central Committee of the Communist Party of China, it was stressed that innovation shall maintain the core position in the overall construction of China's modernization. Since the implementation of China's National Strategy of Innovation-driven Development, Chinese enterprises' innovation capability has been playing an increasingly strong advantage in domestic and international competition. The 2020 China Innovation Index released by the National Bureau of Statistics of China shows that, in 2019, the R&D expenditure of Chinese enterprises reached RMB 169.218 billion, an increase of 11.1% compared with 2018; among industrial enterprises above designated size (annual main business revenue above RMB 20 million), 188,000 of them have carried out technological innovation activities, accounting for 49.6% of the total [2]. Technological innovation brings competitive advantages, as well as changes in the revenue model of enterprises, and is an important guarantee for their success as well as sustainable development.

Enterprise technological innovation can improve production efficiency, which is closely related to enterprise innovation output. Enterprises' innovation output generally refers to the comprehensive evaluation of the efficiency and effectiveness of enterprise technological innovation, and is achieved by the innovation subject through a series of innovative and inventive activities [3]. There are several key factors that may affect the enterprises' innovation output. First, factor input. The resource-based view (RBV) argues that an enterprise is a pool of various types of resources, and their innovative behaviors are related to their ability to acquire resources for R&D inputs. Factor inputs are the most basic, important and direct influences for an enterprise's innovative activities, with capital and labor being the backbone of an enterprise. R&D capital inputs not only affect R&D activities, but also determine the enterprise's R&D strategy choice [4]. Labor, on the other hand, is the factor that has impacts on the stages of research, development and transformation. Second, the cooperative and competitive relationships between enterprises. Considering the increasingly fierce competition among enterprises in a knowledge-intensive environment, enterprises will continuously co-exist in an open and inclusive background and become closely connected with other external entities. More and more competing enterprises will carry out collaborative innovation activities and form cooperative alliances [5] by developing co-opetition strategies and forming co-opetition networks [6]. Hence, the competitive-cooperative relationship among enterprises has an important impact on the innovation output. In addition, the business environment, consisting of policies, infrastructure development and legal environment, etc., also influences enterprises' innovation strategies, and further, their innovation output [7].

Despite the rich findings, previous studies mostly focus on the influence of a single factor or a class of factors on the innovation output of enterprises, but rarely did quantitative evaluation on the innovation output by establishing a comprehensive indicator system. Insomuch as this, this paper analyzes the influencing mechanism of three types of factors, namely, factor input, competitive cooperation relationship and environment, on the innovation output of enterprises. The authors used complex networks to model the co-opetition relationship among enterprises, integrate the factor input and the external environment, and quantitatively analyze the degree of influence of different types of factors on the innovation output of enterprises. What's more, since patents are the concrete manifestation of new technologies, new products and new processes of enterprises, and are widely used as indicators to measure the innovation output of enterprises [8], this paper adopts the number of patent applications as a measurement. The purpose of this paper is to clarify the influence mechanism of different indicators on enterprise innovation output, and then provide relevant suggestions for improving enterprise innovation ability. Therefore, this paper proposes a comprehensive indicator system to evaluate the innovation output of enterprises, and then constructs the multi-influence factor model of enterprise innovation

output with the solution method presented. This research is important for identifying and measuring the influencing factors of enterprise innovation output and quantitatively exploring the influencing mechanism of different factors. With a clear understanding of the influencing mechanism, enterprises can better allocate resources, give full play to their competitive advantages, and continuously carry out technological innovation, thus further upgrading their own innovation output.

This paper is structured as follows: Section 2 reviews the previous literature on the factors influencing enterprise innovation output; Section 3 develops the indicator system of the influencing factors for enterprise innovation output; Section 4 constructs the multi-influence factor model of enterprise innovation output and provides the solution method; Section 5 shows the measurement results and discussion; and Section 6 presents the conclusions and recommendations.

## 2. Literature Review

Innovation behavior generally refers to the behavior that is guided by proposing opinions that are different from the conventional ones, while using the existing knowledge and resources to improve or create new things in a specific environment (including products, methods, elements, paths and environment) at the same time, and can obtain certain beneficial effects [9]. Innovation is also defined as the introduction of the new, novel or reformed [10]. For enterprises, their innovation ability is mainly reflected in the scientific research, technology development and production practice to use the existing foundation, and constantly provide the ability of new theories, new ideas and new methods to promote the development of the above activities. The innovation output of enterprises is its achievements through a series of innovation behaviors and invention activities [3]. The innovation output of enterprises is influenced by several types of factors. The authors summarize previous literature from three aspects, namely, the factor input, the competitive-cooperative relationship between enterprises, and the market environment in which enterprises are located, and explore the influence mechanisms of the above factors on the innovation output of enterprises.

### 2.1. Enterprise Factor Input

Enterprise factor input, as an important internal factor affecting the innovation output of enterprises, is the foundation of enterprise innovation. Enterprise factor input mainly includes the capital factor, labor factor and organizational factor. In the enterprise innovative activities, the innovation behavior of enterprises is mainly manifested in the ability to obtain resources for R&D inputs. First, financial resources are the main driver of innovation [11]. Pegkas [12] examined the relationship between innovation and R&D expenditure in EU countries in the time span from 1995 to 2014. The empirical results show a positive relationship between innovation and R&D, with 10% increase in R&D expenditure leading to 5.8% increase in innovation output. In the study on the relationship between R&D and innovation investment and firm productivity of 14,178 SMEs in South Korea from 2014 to 2018, Bong applied the multiple regression analysis and three-stage least squares method, and found that R&D investment has a positive and significant impact on enterprises' innovation output [13]. Secondly, the labor factor is also a key component of enterprise innovation factor input. People, as the agent of innovation behavior, play a crucial role in the innovation process. It was found that the knowledge, innovation ability and creativity of the R&D personnel are essential elements for the innovative environment in an enterprise [14]. In addition, the number of R&D personnel also has a significant impact on innovation outputs. Rosenbloom [15] studied the relationship between the federal R&D funding received by industry-academia research groups in the United States and the number of postdoctoral personnel employed, and found a significant positive correlation between the two, which means the larger number of postdoctoral researchers leads to more research funding and thus increased innovation output. The internal culture or environment of an enterprise, including organizational structure, working culture and ad-

ministrative processes, will affect its innovation. As the platform for innovation, enterprise organization can coordinate works of different departments and motivate employees, thus contributing to the innovation output. Related studies have also shown that organizational learning can facilitate new area investment and creative technologies [16].

### 2.2. Co-Opetition Relationship between Enterprises

Competitive and cooperative relationships between enterprises in the market can influence the resource allocation process and product production efficiency, which in turn affects the innovative output of enterprises. From the competitive point of view, an enterprise's success greatly depends on the competitiveness of its products and services in rapidly changing market conditions [17]. The competitive market theory suggests that the larger the number of enterprises in an industry, the more intense the market competition will be, which will, to a large extent, affect the market position of core enterprises, as well as the interests of member enterprises [18]. Market competition motivates enterprises to invest more innovation resources. In the face of external competition, enterprises will give full play to their competitive advantages and enhance synergistic cooperation with members within the system to resist competitive risks [19]. For example, Mi found that fiercer market competition in the banking industry will expand enterprises' access to external finance and reduce the cost of financing [20]. Sheikh [21] argued that in a competitive market environment, shareholders are more inclined to delegate more power to corporate managers, who are more likely to use their talents to promote innovation. From a cooperative perspective, cooperative innovation is a crucial part of a country's innovation system [22], and enterprises are one of the key drivers of cooperative innovation. As technological innovation becomes more sophisticated, enterprises can hardly have all the necessitated capabilities to innovate on their own [23]; even for highly competitive ones, it will be conducive for their innovative output to collaborate with those with complementary and innovative resources. By teaming up with each other, enterprises can not only acquire the needed instruments and technical equipment, but also share the expertise, resources and capabilities [24]. As the source of innovation, active collaboration has a significant positive effect on an enterprise's sustainable innovation output [25]. In the context of win–win cooperation, an open innovation platform came into being. With the help of the open innovation platform, the company can carry out cooperative activities more effectively [26,27]. Research showed that the open innovation platform supported knowledge joint creation and knowledge restructuring, especially in high-tech enterprises; the use of the open innovation platform could accelerate the internal innovation process and knowledge outflow, so as to expand the market scale of enterprises and enhance the strategic advantage of enterprises [10].

The co-opetition relationship between enterprises can be displayed via networks. As an enterprise become more connected with other enterprises, universities and research institutions in the industry, the competitive and cooperative relationships between them change from a simple linear way to a complex network [28]. On one hand, competitive networks prompt enterprises to make changes according to their competitors and gain competitive advantages for sustainable development, by sharing information about the competitive pressure among network members and the progress made by their competitors. On the other hand, cooperative network among innovation actors is formed through knowledge spillover and sharing, which can share investment risks, reduce R&D and transaction costs [29], and increase the innovation output of the partners [30]. For enterprises, the co-opetition network and their position in the network can impact their innovation output. It has been suggested that in different organizational settings, the pattern in which the network position of an innovative organization influences its innovation output may vary depending on the network structure and composition [31]. Guo analyzed the influence of network positions of 8727 ICT firms in the cooperation network on their innovation output in the time span from 2002 to 2014 [32]. The results showed an inverted U-shaped relationship between the degree centrality of enterprises and their innovation output in industrial cooperation networks. Similarly, Erik concluded that the relationship between a

firm's capabilities (position) in the network and its innovation performance is inextricably linked [33]. A superior position in a co-opetition network will allow enterprises access to more resources and huge advantages in knowledge transfer and diffusion, which can promote innovative activities and increase innovation output of enterprises.

*2.3. Market Environment*

Enterprise innovative activities are the process of converting internal knowledge and technology into actual output by gathering labor and capital in the organizational structure, which cannot be done without a sound market environment [34]. The market environment mainly includes the political, infrastructural and legal factors. In the political aspect, government subsidies can ease enterprises' needs for financing and, to some extent, alleviate the problems of insufficient R&D investment and high costs [35]. Wang applied an improved two-stage network data envelopment analysis (DEA) method to study the innovation resource allocation efficiency of 58 integrated civil-military enterprises in China, and the results of the study showed that government support had a significantly positive impact on the innovation resource development and overall efficiency [36]. In addition, a combination of government policies for both tax incentives and R&D subsidies can also help with enterprises' R&D investment and innovation output [37]. At the infrastructural level, factors such as science and technology infrastructure (STI) inputs and incubators have their influences. In a survey on 75 academics from six private universities in Iraq, in terms of STI investment indicator, Jabbouri [38] found that STI investment in the information technology industry contributes a lot to technological innovation. Public financial support also helps increase the number of STI investments and the odds of technological innovation. As for incubators, they support enterprises to accelerate their growth or innovation by providing a range of relevant resources and services [39]. From the perspective of resources, Chen [40] analyzed the impact of incubators on the innovation output of 122 new firms with regression models, which shows a positive impact. Fernández [41] pointed out that in an open and innovative environment, incubators provide services to enterprises, which is an innovative and dynamic process, giving rise to the interplay of factors in the entrepreneurial ecosystem. In the legal perspective, a fair and established legal environment is vital for the sound and sustainable development of enterprises. In particular, the emphasis on the protection of intellectual property rights can significantly contribute to the increase in the number of corporate patent applications [42]. Andreea Barbu [43] used multiple regression models to study the effect of intellectual property rights on enterprise innovation, and the results showed that the protection for patent and trademark rights have a significant positive effect on the enterprise innovation output. Intellectual property right (IPR) protection allows firms to transform their innovative potential and creativity into market value, profits and productivity, thus contributing to the enhanced innovation output. However, IPR should be protected in the proper way, in that excessive protection will lead to technological monopoly of large companies, causing SMEs high costs for technological innovation and reduced innovation output [44].

To sum up, in the studies of the influencing factors of enterprise innovation output, previous research separately focused on enterprise factor input, enterprise co-opetition relationship, and the market environment that surrounds the enterprises, all with well-grounded results. However, most of the works had only one focal point, without having a holistic view and integrated analysis on the three major influencing factors. To fill that gap, this paper first constructs an indicator system to influence the innovation output of enterprises by integrating the three influencing factors, namely, factor input, enterprise coopetition and market environment; then, the relevant data of enterprises in Zhongguan-cun Chaoyang Sub-park and Fengtai Sub-park in Beijing, China, are analyzed as the sample of data. Based on spike-and-slab sparse function and stepwise regression model, this paper proposes and gives solutions to the quantitative model of enterprise innovation output and the influencing factors, calculates the regression coefficient and elasticity coefficient of each factor, and explores the influence of different factors on enterprise innovation output.

This research will enrich the literature related to the influencing factors of enterprise innovation output, and lay a theoretical foundation for the refined study of the indicator system for the influencing factors of enterprise innovation output. The results also give guidance to enterprises to scientifically measure and improve their innovation output and reasonably make plans and allocate resources.

## 3. Modelling

### 3.1. Research Objects

This paper takes 562 high-tech enterprises in Chaoyang Sub-park and Fengtai Sub-Park of Zhongguancun (Z-Park) in Beijing, China, as the research objects to study the factors affecting enterprise innovation output. This paper uses relevant data from 2015 to 2016, provided by Beijing Z-park Administrative Committee. Referring to the literature related to the traditional indicator system, this paper focuses on three dimensions, namely, enterprise factor input, enterprise co-opetition relationship, and market environment, to construct an indicator system of influencing factors for enterprise innovation output. The selected indicators and data sources are elaborated as follows:

### 3.2. Indicators Selection and Data Sources of Enterprise Innovation Output

3.2.1. Enterprise Factor Input

Enterprise factor input is the internal influencing factor of enterprise innovation activities. In this paper, capital input and labor (talent) input are mainly studied. First of all, capital is essential for enterprises to carry out innovation; investing capital in innovation activities can both expand enterprises' access to R&D resources and help them attract and retain more research talents. This research mainly considers the following two capital input indicators: (1) the total internal expenditure on science and technology activities, which reflects the level of enterprises' investment directly used for their own innovative R&D activities; and (2) the fixed assets used for science and technology activities in the current year, which reflects enterprises' investment in building and acquiring fixed assets for innovative science and technology activities in the current year. Secondly, labor (talent) is the root of an enterprise's innovation; knowledge, innovation ability and creativity of research personnel constitute an enterprise's internal innovation environment. The labor investment indicator refers to the ratio of the number of scientific and technological personnel to the number of employees at the end of the working period, which reflects how much an enterprise invest in human resources for scientific and technological activities, calculated as: the number of scientific and technological personnel/the number of employees at the end of the working period × 100%. All the aforementioned indicator-related data are from Beijing Z-park Administrative Committee.

3.2.2. Environmental Factor

Environmental factor is the influencing factor for enterprise innovation activities from outside the enterprise. An enterprises' innovation activities are the process of converting the internal knowledge and technology into actual output by gathering people, money and materials in the organizational structure, which cannot be done without the support from the external environment. Enterprise innovation output can be influenced by the economic condition of the industrial park where the enterprise is located, innovation infrastructure and policy support from the government. In this paper, environmental factors mainly include: (1) regional GDP, which reflects the overall economic strength and development stage of a region; (2) the amount of investment in fixed assets, which refers to a region's investment in infrastructure construction, renewal, expansion and repair, and other fixed assets and reflects the volume of regional investment in fixed assets in order for science and technology innovation; (3) the amount of foreign capital actually utilized, which can showcase how open the region is to the outside world, as well as the extent to which the region uses foreign capital to promote innovation; (4) the number of incubators, which can provide entrepreneurial support to entrepreneurs, reflecting how active the region is

involved in innovation and entrepreneurship; (5) investment in innovation culture, which refers to the financial expenditure on the development of culture and arts and the spread of scientific knowledge; (6) government investment in science and technology, which refers to the financial expenditure for supporting scientific research; and (7) the number of newly emerged enterprises, which refers to the number of newly registered enterprises in the year, reflecting the market vitality of the region. All the aforementioned data are from Beijing Z-park Administrative Committee.

### 3.2.3. Co-Opetition Relationship between Enterprises

- Step 1. Constructing the Co-opetition Network of Enterprises
- Step 1-1. Patent Cooperation Network

In the context of economic globalization, enterprise innovation needs more comprehensive elements support, presenting the urgency for technological exchanges and cooperation among companies. Cooperation among enterprises based on complementary resources have become the mainstay in enterprise innovation, and the correlation of technology diffusion between enterprises significantly influences the innovation output of enterprises. In this paper, we construct a Patent Cooperation Network (PCN), based on the enterprises' joint patent application, to study the impact of technological cooperation between enterprises on their innovation output. In this network, patent-applying enterprises are presented as nodes, and between the enterprises that jointly apply for a certain patent, there exist the connecting edges whose weight reflects the level of connection between enterprises. The node set and the edge set thus constitute a weighted, undirected graph $G = (V, E, W)$. Details are elaborated as follows:

(1) The node set consists of companies that jointly apply for patents in Chaoyang Park and Fengtai Park, denoted as $V = \{\overleftrightarrow{v_i}\}$, where $i = \{1, 2, \cdots, n\}$.

(2) The edge set $\overleftrightarrow{E}$ is composed of edges representing the technological connection in the joint patent application between enterprises. If enterprise $v_i$ and enterprise $v_j$ jointly apply for a patent, there will be an edge $e_{ij}$ and $e_{ij} = 1$; if there is no patent cooperation between $v_i$ and $v_j$, there will be no edge, so, $e_{ij} = 0$. Considering the symmetric nature of the cooperative relationship between the enterprise and the enterprise, i.e., $e_{ij} = e_{ji}$. Here, $i, j \in \{1, 2, \cdots, N\}$.

(3) Usually, we use the distance matrix $W = \{w_{ij}\}$ to describe the entity structure of the network, where $w_{ij}$ represents the weight between node $i$ and node $j$. If there is no patent cooperation between $v_i$ and $v_j$, then $w_{ij} = 0$; one joint patent application will be denoted as $w_{ij} = 1$, two joint applications as $w_{ij} = 2$, and so forth.

Following the abovementioned constructing principles of the model, we obtain a PCN model with 53 nodes and 31 edges, based on the joint patent application data of 571 enterprises in Chaoyang Park and Fengtai Park of Z-park in Beijing between 2015 and 2016. The PCN topology of the enterprises is shown in Figure 1.

- Step 1-2. Capital Co-opetition Network

Whether an enterprise's innovation output is active or not is closely related to its capital input. Financial investment underpins an enterprise's innovation activities, increases the amount and intensity of the enterprise's investment in innovation R&D, and elevates its innovation capability. In this paper, we construct the Capital Co-opetition Network (CCN) based on the investment and financing between enterprises, to analyze the cooperation relationship between investing enterprises and the competition relationship between financing enterprises, and study the influence of the investment and financing relationship between enterprises on the innovation output of enterprises. In constructing the network, we take the investing enterprises and financing enterprises as nodes, and their relationships as the connected edges, whose weights indicate the investment amount, thereby obtaining the directed weighted graph of the CCN model. The detailed modeling principles are as follows:

(1) The node set denoted as O $=\{o_{ij}\}$ consists of financing enterprises, and the node set P $=\{p_{ij}\}$ consists of investing enterprises, where $i, j \in \{1, 2, \cdots, N\}$.

(2) The set of edges consists of the edges that represent the investment-financing relationship between enterprises. If there is an investment activity from enterprise $o_i$ to enterprise $p_j$, there will be an edge $e_{ij}$ and $e_{ij} = 1$; if there is no investment activity from $o_i$ to $p_j$, there will be no edge, so, $e_{ij} = 0$. Since the investment and financing relationships between $o_i$ and $p_j$ are not always pairwise, it might appear that $e_{ij} \neq e_{ji}$. Here, $i, j \in \{1, 2, \cdots, N\}$.

(3) We used distance matrix $W = \{w_{ij}\}$ to describe the entity structure of the network, where $w_{ij}$ represents the weights between nodes. If there is no investment activities from $o_i$ to $p_j$, then $w_{ij} = 0$; if there is an investment activity from $o_i$ to $p_j$, then $w_{ij} = 1$; if there are two investment activities from $o_i$ to $p_j$, then $w_{ij} = 2$, and so on.

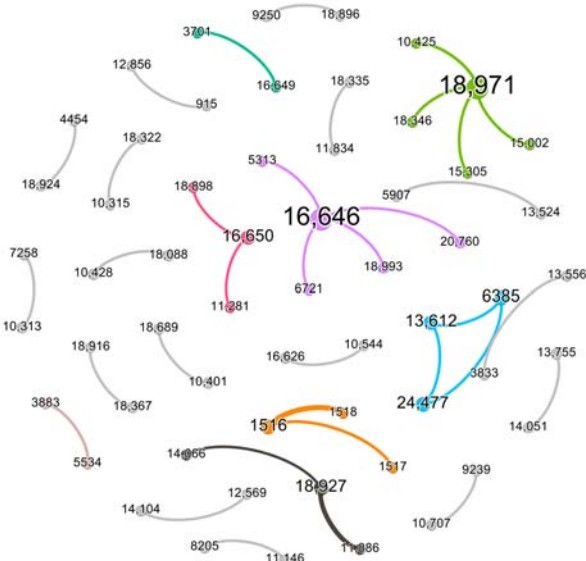

**Figure 1.** PCN Topology of enterprises in Chaoyang Park and Fengtai Park. Enterprises connected by the same color line indicate that they have patent cooperative relationship. Data source: Z-Park Management Committee.

Based on the model construction principles, we obtained the CCN model of 397 nodes and 262 edges with the investment and financing data of 571 enterprises in Chaoyang Park and Fengtai Park of Z-Park in Beijing between 2015 and 2016. Figure 2 shows the topology diagram of those enterprises in the CCN:

- Step 1-3. Product and Service Competition Network

The competitive environment faced by the products and services of an enterprise influences enterprise innovation output from the outside through first affecting the enterprise's strategies. In this part, a Product and Service Competition Network (PSCN) has been constructed to analyze the impact of inter-enterprise product and service competition on enterprise innovation output. In this network, nodes denote the enterprises in Chaoyang Park and Fengtai Park, and edges mean the two enterprises have competing products or services. The detailed constructing principles of this undirected and unweighted PSCN model $G = (V, E)$ are elaborated as follows:

(1) The node set $V = \{v_{ij}\}$ is comprised of the enterprises in Chaoyang Park and Fengtai Park, where $i, j \in \{1, 2, \cdots, N\}$.

(2) The edge set $E$ represents the competition among enterprises with competing products or services. If enterprise $v_i$ and enterprise $v_j$ compete with similar products or services, there will be an edge, i.e., $e_{ij} = 1$; if $v_i$ is not in competition with $v_j$, there will be

no edge, i.e., $e_{ij} = 0$; if the competitive relation between $v_i$ and $v_j$ is bilateral, then $e_{ij} = e_{ji}$. Here, $i, j \in \{1, 2, \cdots, N\}$.

(3) We used distance matrix $A = \{e_{ij}\}$ to present the real structure of the network. If there is no competition between $v_i$ and $v_j$, $e_{ij} = 0$; otherwise, $e_{ij} = 1$.

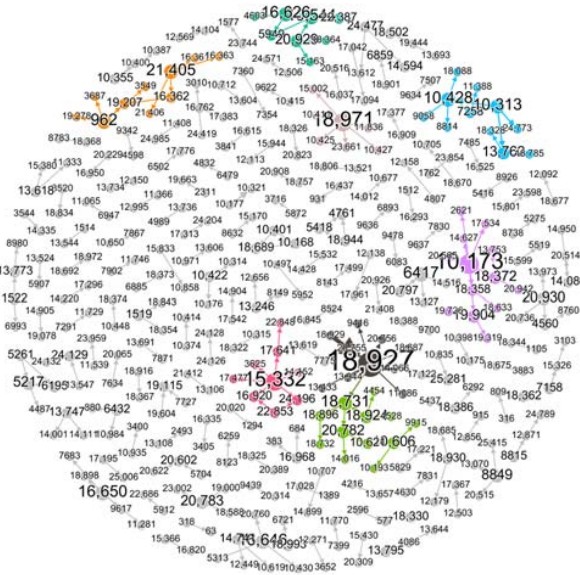

**Figure 2.** CCN Topology of enterprises in Chaoyang Park and Fengtai Park. Enterprises connected by the same color lines indicate that they have capital co-opetition relationship. Data source: List of High-tech Enterprises in Z-park formulated by Z-park Management Committee and https://www.tianyancha.com/ (accessed on 12 December 2015).

Based on these principles, we constructed a PSCN model of 306 nodes and 216 edges according to the produce and service competition data of 571 enterprises in Chaoyang Park and Fengtai Park of Z-Park between 2015 and 2016. Figure 3 illustrates the PSCN topology of the enterprises:

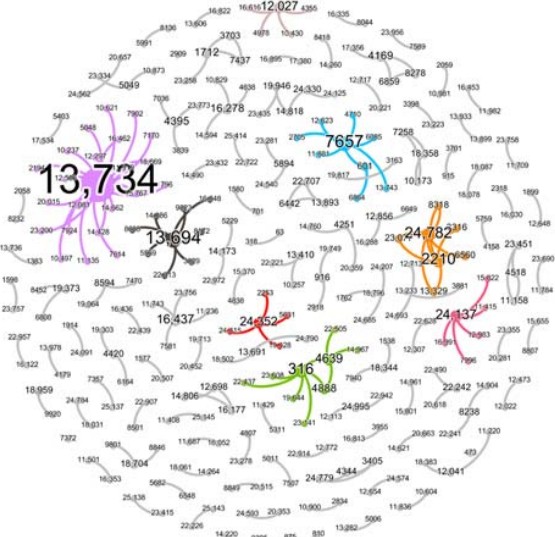

**Figure 3.** PSCN Topology of enterprises in Chaoyang Park and Fengtai Park. Enterprises connected by the same color lines indicate that they have product and service competition relationship. Data source: List of High-tech Enterprises in Z-park formulated by Z-park Management Committee and https://www.tianyancha.com/ (accessed on 12 December 2015).

- Step 2. Extract network structural indicators

The technological collaboration, capital operation and market competition of enterprises all reflect the interplay between enterprises. As knowledge sharing and resource sharing between enterprises become more and more frequent, the market competition between enterprises becomes more and more intense, which will exert greater influence in enterprises' innovation output. For instance, joint patent application, investment and financing, and product and service competition, etc. may all shape the innovation output of enterprises. In this paper, therefore, we use the network structure indicators from the co-opetition network of enterprises to study the influence of enterprise co-opetition on their innovation output. The network is built up through the process detailed in the animation (Online Resource).

For the structural indicators of the enterprise co-opetition relationship network, this paper will describe the innovation capability of enterprises in four dimensions: innovation breadth, innovation intensity, innovation density and innovation depth. The node degree $K(i)$ is to quantify the innovation breadth, calculated as in Equation (1):

$$K(i) = \sum\nolimits_{j \in \tau(i)} e_{ij} \tag{1}$$

Node weight $S(i)$, unit weight $Z(i)$ and disparity in edge weight $Y(i)$ are used to quantify the innovation intensity, which is calculated as in Equation (2):

$$S(i) = \sum\nolimits_{j \in \tau(i)} w_{ij}, \ Z(i) = \frac{S(i)}{K(i)}, \ Y(i) = \sum\nolimits_{j=1}^{N} \left[ \frac{w_{ij}}{S(i)} \right]^2 \tag{2}$$

The clustering coefficient of nodes $C(i)$ and the weighted clustering coefficient of nodes $C^w(i)$ are used to quantify the innovation density, which is calculated as in Equation (3):

$$C(i) = \frac{2A(i)}{K(i)(K(i)-1)}, \ C^w(i) = \frac{\sum_{j,k \in G} w_{ijk} \, e_{ij} \, e_{jk} \, e_{ki}}{K(i)(K(i)-1)} \tag{3}$$

where $A(i)$ refers to the actual number of edges of adjacent nodes, and $w_{ijk}$ represents the influencing factor of the weight of the triangle with node $i$ at the core.

The innovation depth of the enterprise is quantified by the nodes' betweenness centrality $C_B(i)$ and closeness centrality $C_C(i)$, which is calculated as in Equation (4):

$$C_B(i) = \sum\nolimits_{i,j,k \in \{1,2,\cdots,N\}} \frac{d_{jk}(i)}{d_{jk}}, \ C_c(i) = \frac{N-1}{\sum_{j=1}^{N} d_{ij}} \tag{4}$$

In the selection of structural indicators of networks, the same structural indicators often have different meanings in different networks. Therefore, we explain the meanings of indicators according to specific networks:

(1) Select the structural indicators of the enterprise patent cooperation network (PCN). The degree of enterprise node $v_i$ refers to the number of enterprises with which a certain enterprise has joint patent applications. The larger the degree, the greater influence the enterprise exerts in the network, the more technical partners the enterprise has. The weight of $v_i$ quantifies the intensity of technological collaboration among enterprises; the unit weight of $v_i$ refers to the quality of the enterprise's innovation technology. The edge weight disparity of $v_i$ describes the dispersion degree of the intensity of technological collaborative relationship between enterprises—higher disparity signifies more significant industry characteristics and more concentrated innovation areas of an enterprise. The betweenness centrality of $v_i$ measures the mediating role of an enterprise in the technology collaboration network—the larger the value, the more important an enterprise is in the network and the higher its status.

(2) Select the structural indicators of the enterprise capital co-opetition network (CCN). The node weight of $v_i$ includes inward node weight and outward node weight. The latter

denotes the intensity of an enterprise's cumulative investment to other enterprises, while the inward node weight reveals the intensity of an enterprise's cumulative financing from external channels. The unit weight of $v_i$ measures the quality of enterprise co-opetition. A higher outward unit weight signifies a higher average level of an enterprise's outward investment; a higher inward unit weight means a higher average level of financing. The edge weight disparity of $v_i$ describes the dispersion degree of enterprises' competitors and partners. The higher the disparity in outward edge weight, the more oriented the enterprise is in its investment activities, while higher disparity in inward edge weight means that the financing source of enterprises in financing activities is relatively single, and the actual financing is from mostly inside its group, with limited sources and less attraction from external financing funds.

(3) Select the structural indicators of the enterprise product and service competition network (PSCN). In this case, the degree of $v_i$ refers to the number of enterprises with which a certain enterprise competes—the larger the degree value, the more competitors it has. The unit weight of $v_i$ quantifies the intensity of competition for the enterprise's products and services. The edge weight disparity of $v_i$ describes the dispersion degree of the intensity of competitive relationship between enterprises in terms of products and services. The clustering coefficients of $v_i$ are a gauge of the cliquishness between enterprises. The higher the clustering coefficients, the more intense the competition between enterprises in the context of industrial cluster. The betweenness centrality of $v_i$ measures the intermediating role of the enterprise in information flow with other enterprises, and the nodes with higher betweenness centrality play a crucial role in the innovation value chain. The closeness centrality of $v_i$ measures the closeness and centrality degree of the enterprise in the network topology. The higher the closeness centrality, the more advantageous a position it has taken.

In summary, a total of 27 indicators are selected as influencing factors for the enterprise innovation output, including 17 indicators of network structure, 3 indicators of factor input and 7 indicators of environment, as shown in Table 1. In the following section, the influencing mechanism of these indicators will be analyzed.

**Table 1.** A Comprehensive Indicator System of Influencing Factors for Enterprise Innovation Output.

| Primary Indicators | Secondary Indicators | Tertiary Indicators |
|---|---|---|
| Network Structure (X1) | Patent Cooperation Network | degree ($x_{i1}^1$), node weight ($x_{i2}^1$), unit weight ($x_{i3}^1$), edge weight disparity ($x_{i4}^1$), betweenness centrality ($x_{i5}^1$), inward node weight ($x_{i6}^1$) |
| | Capital Co-opetition Network | outward node weight ($x_{i7}^1$), inward unit weight ($x_{i8}^1$), inward unit weight ($x_{i9}^1$), disparity in inward edge weight ($x_{i10}^1$), disparity in outward edge weight ($x_{i11}^1$) |
| | Product & Service Competition Network | degree ($x_{i12}^1$), unit weight ($x_{i13}^1$), edge weight disparity ($x_{i14}^1$), clustering coefficient ($x_{i15}^1$), closeness centrality ($x_{i16}^1$), betweenness centrality ($x_{i17}^1$) |
| Factor Input (X2) | Capital | internal S&T budget ($x_{i18}^2$), fixed asset investment in S&T activities ($x_{i19}^2$) |
| | Labor | S&T staff/all working staff ($x_{i20}^2$) |
| Environmental Factor (X3) | Environmental Factor | regional GDP ($x_{i21}^3$), fixed asset investment amount ($x_{i22}^3$), foreign investment in actual use ($x_{i23}^3$), number of incubators ($x_{i24}^3$), investment in innovative culture ($x_{i25}^3$), government S&T investment ($x_{i26}^3$), number of newly emerged enterprises ($x_{i27}^3$) |

Note: The data of enterprise factor input and environmental factors are the average of annual data in 2015 and 2016.

## 4. Methodology

In this paper, we study the correlation between the innovation output of different enterprises and the influencing factors. We first construct a quantitative model between enterprise innovation output and each influencing factor, give solutions to the proposed model based on spike-and-slab sparse function and stepwise regression model, and then measure the regression coefficient and elastic coefficient of each factor indicator to analyze the influence of different factors on enterprise innovation output.

First, the functional relationship between enterprise innovation output and each influencing factor can be shown as in Equation (5):

$$y_i = \alpha + X\beta + \varepsilon_i \tag{5}$$

where $\alpha$ denotes the intercept term; $\varepsilon_i$ denotes the residual term; $y_i$ denotes the innovation output of each enterprise; $X = (x_{ij}^l)$ denotes the 27 L2 indicators of 562 firms, in which $l = 1, 2, 3$, $i = 1, 2, \cdots, 562$, $j = 1, 2, \cdots, 27$; and $\beta$ denotes the regression coefficients of each variable in the regression equation.

Next, the variable selection model is constructed based on the spike-and-slab sparse function and stepwise regression to solve Equation (5). Considering the multicollinearity in the large number of independent variables in Equation (5), OLS cannot accurately measure the influence coefficients of the independent variables on the dependent variable. Besides, unless the variable has no effect, it is hoped that the maximum amount of the influencing factors can be kept in the model. The specific solution steps are as follows:

Step 1: construct the variable selection model. Independent variable $X$ is randomly sampled, based on the spike-and-slab sparse function. Denote the coefficient of the independent variable $X$ as the column vector $\beta = (\beta_j)$, according to which $\gamma = (\gamma_j)$, where when $\beta_j = 0$, $\gamma_j = 0$ and when $\beta_j \neq 0$, $\gamma_j = 1$. It is usually possible to construct $\gamma$, based on the Bernoulli distribution, as shown in Equation (6):

$$\gamma \sim p_j{}^{\gamma_j} (1 - p_j)^{1-\gamma_j}, \; j = 1, 2, \cdots, 27 \tag{6}$$

where $p_j$ can be determined according to the ideal number of independent variables $m$ in the stepwise regression process. $p_j = \frac{m}{n}$, where $n$ denotes the total number of independent variables ($n = 27$, m = 11 in this paper). Then, after a priori $\gamma$ is sampled according to Equation (6), $\gamma_j = 1$, so $\beta_j \neq 0$; thus, the corresponding variable $x_{ij}^l$ are selected and denoted as the set $X_\gamma$, which is the set of independent variables for the current stepwise regression process.

Step 2: construct the stepwise regression model. The regression of the dependent variable $y_i$ is performed using $X_\gamma$, to obtain the regression results at the specified significance level (0.05 in this paper). Ensure model convergence with repeated sampling (10,000 times in this paper). The regression parameter of the $r$th stepwise regression is denoted as $\varphi^{(r)} = (\alpha, \beta)^{(r)}$, and a series of fitted results $(\varphi^{(r)})$ can be obtained. The means of all regression coefficients are used as the final estimated coefficients of the independent variables, i.e., $\overline{\varphi} = \frac{\sum_{r=1}^{R} \varphi^{(r)}}{R}$; thus, the model of the relationship between innovation output and each influencing factor is shown in Equation (7):

$$\hat{y}_{ij} = \overline{\alpha} + \overline{\beta}_1 x_{i1}^1 + \overline{\beta}_2 x_{i2}^1 + \cdots + \overline{\beta}_{27} x_{i27}^3. \tag{7}$$

Step 3: Calculate the elasticity of firm innovation output. In order to eliminate the interference of different factor magnitudes on the influence of the dependent variable, we calculate the elasticity coefficients corresponding to different indicators to represent the influence of different factors on enterprise innovation output. Using Equation (8),

we calculate the elasticity magnitude of each influencing factor on innovation output of different enterprises:

$$D_{dj} = \frac{\left(\frac{\Delta x_{ij}^l}{x_{ij}^l}\right)}{\left(\frac{\Delta y_i}{y_i}\right)} \qquad (8)$$

## 5. Results and Conclusions

### 5.1. Analysis on the Influencing Factors of Enterprise Innovation Output

In terms of innovation, the number of patent applications is a key gauge of the intellectual property owned by an enterprise. Widely used as an indicator of an enterprise's innovation output, patents manifest an enterprise's new technology, new products and new techniques. Based on this, in this paper, we use the number of patent applications to characterize the level of enterprise innovation output. To be specific, the number of patent applications is taken as the dependent variable in Equation (7), with the premise that the error term fluctuates around 0 and can pass the stationary test; the goodness-of-fit and the fitting results are shown in Table 2. From Figure 4, which gives the true and fitted values of patent application amounts, it can be seen that the fitted values in Chaoyang Sub-park and Fengtai Sub-park tend to be consistent with the actual patent application numbers, which, to a certain extent, indicates relatively good-fitting results and sound robustness in the influencing factor model we have constructed.

**Table 2.** Estimated Parameters, Regression Coefficients and Elastic Coefficients of the Multiple Linear Regression Model.

| Parameters | Regression Coefficients | Elastic Coefficients | Parameters | Regression Coefficients | Elastic Coefficients |
|---|---|---|---|---|---|
| $R^2$ | 0.93 | / | $\beta_{14}$ | −5.2255 | −1.1294 |
| $\alpha$ | 15.8007 | / | $\beta_{15}$ | −17.6133 | −5.9989 |
| $\beta_1$ | 9.0413 | 2.8321 | $\beta_{16}$ | −17.8252 | −2.3659 |
| $\beta_2$ | 1.4697 | 1.2190 | $\beta_{17}$ | −0.0001 | −5.3482 |
| $\beta_3$ | 8.4653 | 3.4107 | $\beta_{18}$ | 0.0001 | 2.8491 |
| $\beta_4$ | −55.0892 | −9.6907 | $\beta_{19}$ | 0.0019 | 4.6694 |
| $\beta_5$ | 0.0214 | 5.0937 | $\beta_{20}$ | 0.3185 | 5.4835 |
| $\beta_6$ | 0.0000 | 0.3868 | $\beta_{21}$ | 0.0003 | 0.4090 |
| $\beta_7$ | 0.0000 | 0.1031 | $\beta_{22}$ | 0.0022 | 1.1327 |
| $\beta_8$ | 0.0002 | 3.8979 | $\beta_{23}$ | 0.0000 | 0.1782 |
| $\beta_9$ | 0.0000 | 0.0360 | $\beta_{24}$ | 0.1018 | 1.0180 |
| $\beta_{10}$ | −11.5322 | −4.8628 | $\beta_{25}$ | 0.0000 | 0.2609 |
| $\beta_{11}$ | −7.0154 | −1.6623 | $\beta_{26}$ | 0.0000 | 0.3618 |
| $\beta_{12}$ | 0.1065 | 0.5091 | $\beta_{27}$ | −9.1994 | −0.3290 |
| $\beta_{13}$ | −26.9930 | −16.5772 | | | |

Note: All regression coefficients pass the *p*-value test at the significance level of 0.05.

All the indicators in Table 2 passed the significance test, with a satisfying goodness-of-fit at 0.93. Overall, 66.7% of the indicators have a positive impact on the number of patent applications of enterprises, and 33.3% have a negative impact.

Factor input indicators reflect the degree of enterprises' internal investment in innovative technologies. As can be seen from Table 2, the increases in internal funding for S&T activities ($x_{i18}^2$), fixed assets for S&T activities ($x_{i19}^2$) and in the proportion of S&T personnel among the employees ($x_{i20}^2$) can all facilitate patent R&D to a certain extent. Environmental factor indicators reflect the economic and social development of the industrial park where the enterprise is located, as well as government's policy support. The study shows that the number of incubators ($x_{i24}^3$) has a relatively significant positive impact on the number of patent applications, indicating that the more prosperous the regional innovation and entrepreneurial activities are, the higher the number of patent applications of enterprises in the region. The larger number of newly registered enterprises ($x_{i27}^3$) in the year, on the other

hand, may bring about stronger market dynamics and yet excessive competition intensity, leading to a reverse inhibiting effect on the R&D activities of enterprises.

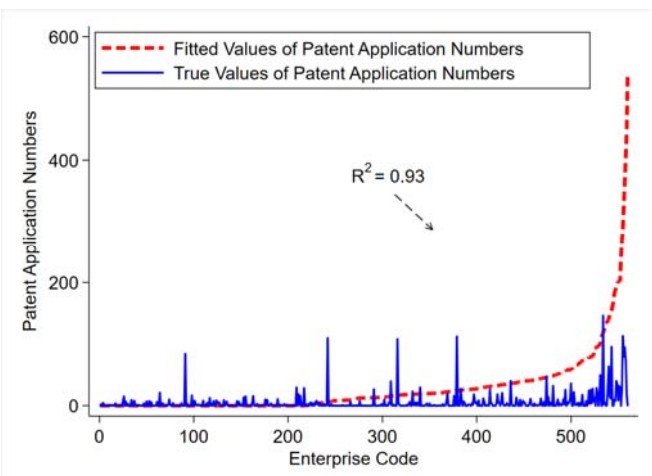

**Figure 4.** True and Fitted Values of Enterprise Patent Application Numbers.

The structural indicators in patent cooperation network reflect the synergistic cooperation relationship generated by enterprises in the process of emerging technology diffusion. As can be seen from Table 2, the degree ($x_{i1}^1$), unit weight ($x_{i3}^1$) and betweenness centrality ($x_{i5}^1$) of enterprise nodes all positively affect the number of patent applications, which indicates that the stronger the enterprise's patent collaboration ability and patent technology advantages, the more important position the enterprise has in the patent cooperation network, and the higher number of patent applications the enterprise possesses. On the contrary, the disparity in the edge weight of patent cooperation relationship between an enterprise and others ($x_{i4}^1$) has a greater negative impact on the number of patent applications, which signifies that the overly concentrated innovation fields will limit the scope of an enterprise's innovation technology to a certain extent and influence its patent R&D. The structural indicators in the capital co-opetition network reflect an enterprise's cooperative relationship with its investing enterprises and the competitive relationship with its financing enterprises. As can be seen from Table 2, the inward unit weight of enterprise nodes has a positive impact on the number of patent applications, which indicates that the higher level of financing an enterprise has, and the more stable the sources of funds, the more patent applications the enterprise will possess. On the contrary, the disparity in inward edge weight ($x_{i10}^1$) and the disparity in outward edge weight ($x_{i11}^1$) have a negative impact on the number of patent applications. Higher disparity in inward edge weight represents limited sources of financing and low attraction to external funds for the enterprises, whose actual financing may be mostly internal group funds. The limited sources of financing increase the heterogeneity of capital co-opetition network, which exposes enterprises to greater risks and reduces its enthusiasm in technology R&D. Higher disparity in outward edge weight reveals the enterprise's investment preference and inclination to make large investments in a few enterprises. Normally, an enterprise can absorb and transfer resources such as technology and knowledge from the subsidiaries it invested, which will provide itself with diverse sources of knowledge and thus a variety of perspectives and ideas, thereby stimulating innovation and facilitating patent R&D. Investing in only a few enterprises is therefore not conducive to the enterprise's access to diverse sources of knowledge, or to their patent R&D.

The structural indicators in the product and service competition networks reflect the competitive relationships among enterprises with similar products and services. As can be seen from Table 2, the unit weight ($x_{i13}^1$), edge weight disparity ($x_{i14}^1$), clustering coefficient ($x_{i15}^1$), closeness centrality ($x_{i16}^1$) and betweenness centrality ($x_{i17}^1$) of enterprise nodes in the product and service competition network all have a negative impact on the number of

patent applications. This implies that the intensity of competition faced by the enterprise's products and services, the intensity of competition between the enterprise and others in the industrial cluster, and the centrality of enterprises in this network all have negative correlations with its innovation output. The market's winner-takes-all mechanism squeezes the enterprise's space for survival in intense competition, further restricting their R&D activities.

In summation, both the internal capital factor investment and the proportion of R&D employees have obvious positive impacts on enterprise innovation output, testifying the crucial role played by capital and talents in innovation activities. Also, the number of technological partners in the process of joint patent application, the intensity of cooperation among enterprises, the enterprise's own technological innovation capabilities and its position in the patent cooperation network positively affects innovation output. This emphasizes that enterprises can seek cooperation with others to exchange and collaborate on knowledge, technology and innovation resources, while enhancing their own innovative capabilities. Besides, the level of financing has a large positive impact on innovation output, yet disparity in both the inward and outward edge weights will limit the innovation output. This indicates that the innovation output of enterprises may prosper with adequate financial support and diverse inputs of financial and knowledge sources, but shrink with fierce product and service competition. In addition, a sound market and policy environment also play a crucial role in innovation output.

### 5.2. Variance Analysis on the Impacts of Primary Indicators on Enterprise Innovation Output

Through measuring with Equation (5) to Equation (7), we can obtain the influence of the three indicators, namely, network structure, factor input and environmental factor on the number of patent applications. In order to better observe the influence of these 3 primary indicators on enterprise innovation output, we selected 346 enterprises with non-negative fitted values of patent applications and arrange them in ascending order, according to the fitted values of patent applications, according to which we can analyze the influence of each primary indicator on enterprise innovation output. The results are shown in Figure 5. Overall, enterprise factor input and the environmental factor have positive impacts on enterprise innovation output, while the network structure indicators have different degrees of influence on the innovation output of different enterprises. Enterprise innovation output is mainly influenced by the network structure and factor input, to a less extent by the environmental factor.

Next, in order to examine the impact of primary indicators on enterprise innovation output, the enterprises are sorted in ascending order of the number of actual patent applications. We categorize the enterprises ranked after 50 in the number of actual patent applications as Group 1, representing the enterprises with a low level of innovation, and the enterprises in the top 50, in terms of the number of actual patent applications, as Group 2, representing enterprises with a high level of innovation. Figure 6 shows the impacts of primary indicators on the innovation output of the two groups of enterprises. According to the results obtained by solving the influencing factor model, we can not only measure the fitting value of the patent application number of enterprises, but also measure the fitting value of three primary indicators. Through the contribution degree of each primary indicator fitting value to the fitting value of the patent application number, we can quantify the specific degree of influence. Overall, the three primary indicators, namely, network structure, factor input and the environmental factor, have different impacts on enterprises at different innovation levels. For those with lower innovation levels (as shown in Figure 6a), innovation output is influenced positively by factor input and the environmental factor, and negatively by network structure factors. Among them, the degree of influence of factor input on enterprise innovation output is about six times higher than that of the environmental factor. For enterprises with higher innovation levels (as shown in Figure 6b), innovation output is influenced positively by network structure and factor input, and the degree of positive influence of the network structure is about 1.6 times as that of the factor

input, and about 6 times as that of the environmental factor; the degree of positive influence of factor input and the environmental factor is lessened.

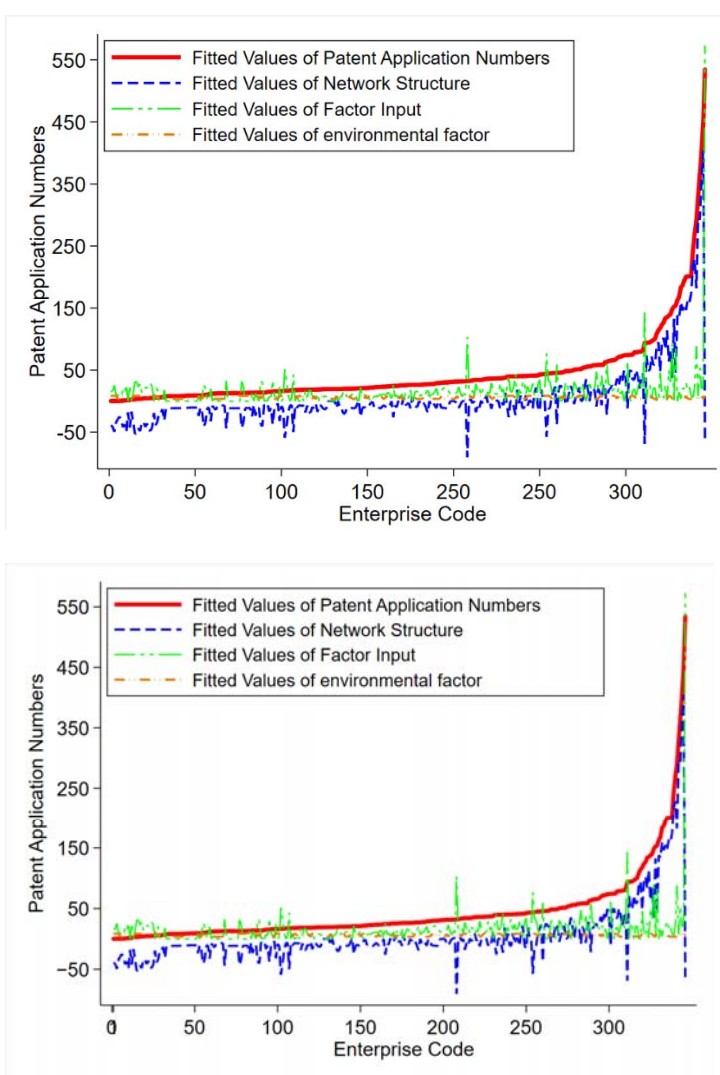

**Figure 5.** Impact of Primary Indicators on Enterprise Innovation Output.

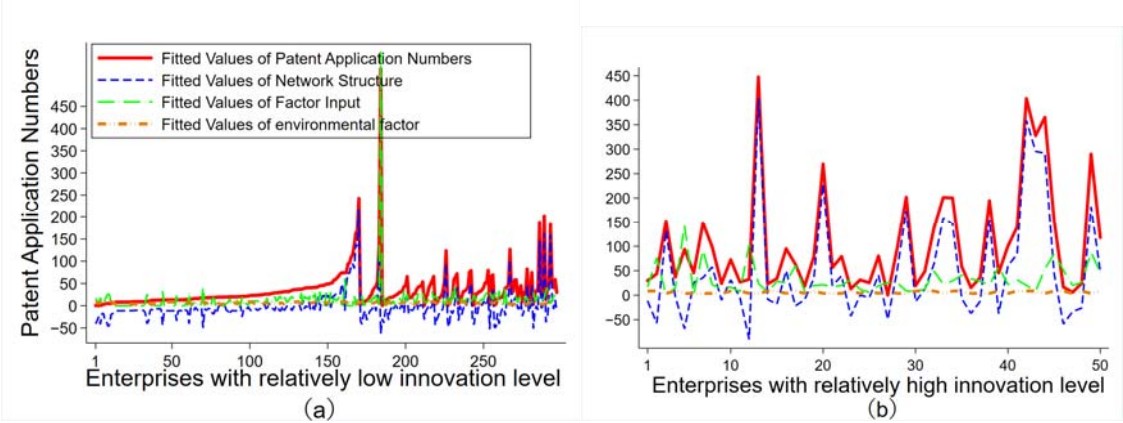

**Figure 6.** Impacts of Primary Indicators on Innovation Output of Enterprises with Different Innovation Levels. (**a**) Impacts of Primary Indicators on Innovation Output of Enterprises with relatively low Innovation Level. (**b**) Impacts of Primary Indicators on Innovation Output of Enterprises with relatively high Innovation Level.

It can thus be concluded that, in order to improve innovation output and maintain their relatively higher innovation level, Group 2 enterprises should put their emphasis on strengthening the advantages of existing enterprise innovation network structure and elevating their positions in the enterprise network, while enhancing factor input and environmental factors. For enterprises with relatively low innovation level, they should pay attention to the optimization of factor inputs and the environmental factor, and also the improvement of their own innovation network structure, in order for better innovation level and output.

The purpose of this study is to provide relevant suggestions for improving the innovation output capacity of enterprises by studying the influence mechanism of indicators on the innovation output of enterprises. Therefore, after quantifying the relationship between enterprise innovation output and its influencing factors, we focused on analyzing those indicators that play an important role in enterprise innovation output and the factors that hinder the improvement of enterprise innovation ability. According to the quantitative results, some indicators have not been discussed in this section.

### 5.3. Variance Analysis on Impacts of Secondary Indicators on Enterprise Innovation Output

5.3.1. Impacts of Secondary Indicators of Network Structure on Enterprise Innovation Output

By measuring with Equation (5) to Equation (7), we can obtain the fitted values of three secondary indicators of network structure, namely, Patent Cooperation Network (PCN), Capital Co-opetition Network (CCN) and Product and Service Competition Network (PSCN). In order to better analyze the impacts of these three secondary indicators on enterprise innovation output, we selected 346 enterprises with non-negative fitted values, in terms of patent applications, and sorted them in the ascending order of the values, according to which we analyzed the influence of the secondary indicators and achieved the results shown in Figure 7. On the whole, PCN has a positive impact on enterprise innovation output, PSCN has a negative impact, and the impact of CCN varies for different enterprises.

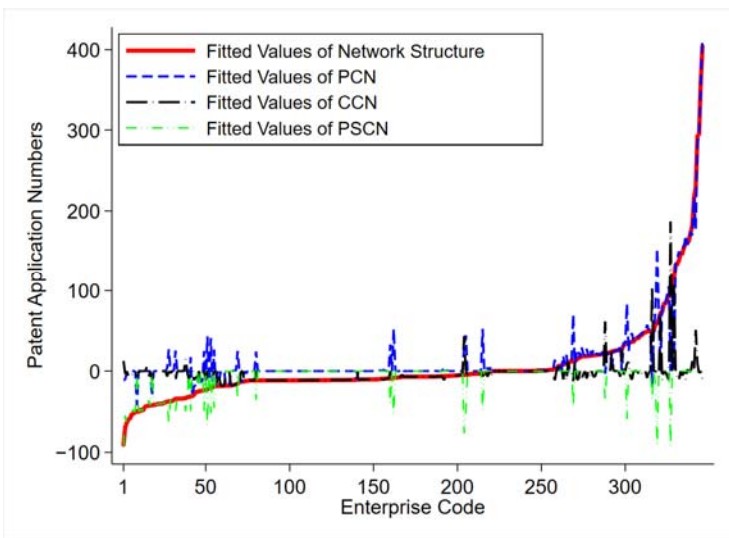

**Figure 7.** Impacts of Secondary Indicators of Network Structure on Enterprise Innovation Output.

Next, in order to analyze the impact of each secondary indicator of network structure on enterprises with different innovation levels, we ranked the enterprises in the ascending order of actual patent application numbers, and divided them into two groups—Group 1 is enterprises ranking after 50 with relatively lower innovation levels, and Group 2 is the top 50 enterprises with relatively higher innovation levels. Figure 8 presents the impacts of the secondary indicators on the innovation output of the two groups of enterprises. For

Group 1 enterprises (as shown in Figure 8a), innovation output is influenced positively by PCN and negatively by CCN and PSCN. The negative influence of PSCN on innovation output is about 1.85 times that of CCN. For Group 2 enterprises (as shown in Figure 8b), innovation output is influenced positively by PCN and negatively by PSCN; CCN also has a negative influence, yet to a lesser degree. The negative impact of PSCN is about 22 times higher than that of CCN. Compared with those with a lower innovation level, the degree of positive influence exerted by PCN surges and the degree of negative influence of PSCN is reduced.

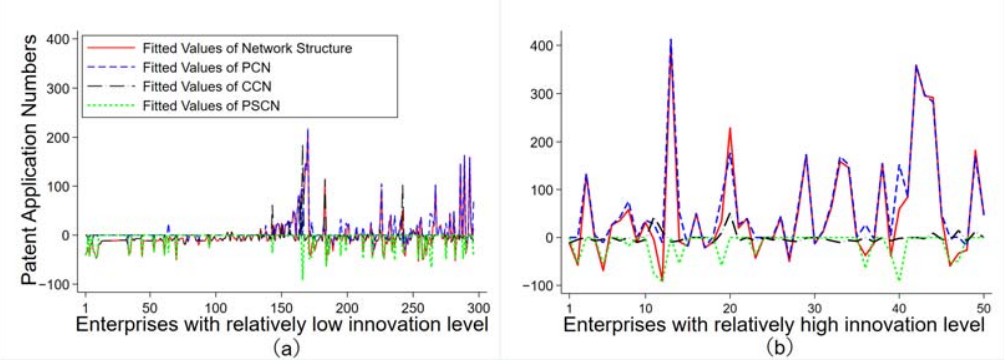

**Figure 8.** Impacts of Indicators of Network Structure on the Innovation Output of Enterprises with Different Innovation Levels. (**a**) Impacts of Indicators of Network Structure on Innovation Output of Enterprises with relatively low Innovation Level. (**b**) Impacts of Indicators of Network Structure on Innovation Output of Enterprises with relatively high Innovation Level.

It can be known from the analysis that PCN plays a more important role in innovation output of enterprises, especially for those with higher innovation level. Hence, when optimizing the existing network structure, enterprises with a higher innovation level should focus on enhancing their positions in PCN by teaming up with more technology partners, to strengthen their voice and create favorable technological conditions for innovation. For enterprises with a relatively low innovation level, their attention shall be paid not only to the optimization of PCN, but also to the elimination of the negative influence in CCN and PSCN.

### 5.3.2. Impacts of Secondary Indicators of Factor Input on Enterprise Innovation Output

To analyze the influence of the two secondary indicators, namely capital factor input and labor factor input, on enterprise innovation output, we sampled 346 enterprises with non-negative fitted values of patent applications and sorted them in ascending order of the fitted values of factor input indicators; the results are shown in Figure 9. In general, both capital factor and labor factor have a positive impact on the innovation output, with the latter having a greater positive impact—1.5 times to the former, according to Figure 9.

Furthermore, in order to examine the impact of each secondary indicator of factor input on enterprises with different innovation levels, we rank the enterprises according to the number of actual patent applications in ascending order—Group 1 has the enterprises ranked after 50 with lower innovation levels, and Group 2 has the enterprises ranked in the top 50 with higher innovation levels. Figure 10 demonstrates the influence of each secondary indicator of factor input on enterprises with different innovation levels. It can be found that for enterprises with lower innovation levels (as shown in Figure 10a), the innovation output is influenced positively by the capital factor input indicator and labor factor input indicator, and the latter exerts greater positive influence, about twice as much as that of capital factor input. This is also true for enterprises with higher innovation level (as shown in Figure 10b), only the positive influence of capital factor is significantly higher, about 1.6 times that of labor factor.

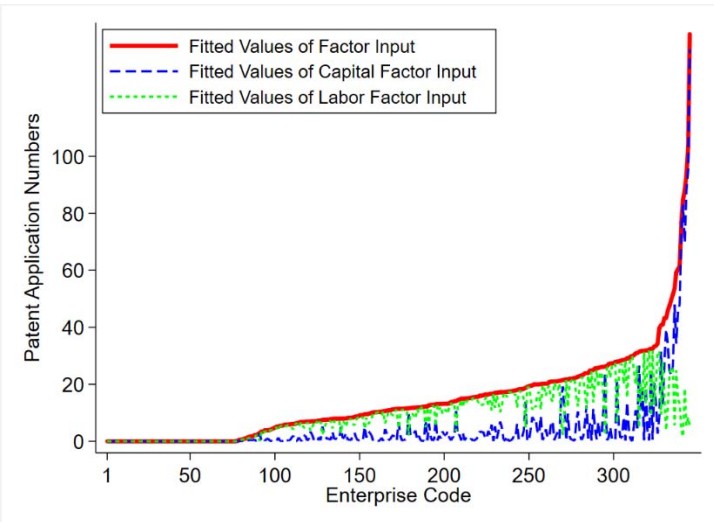

**Figure 9.** Impact of Factor Input Indicators on Enterprise Innovation Output. Note: In order to better present the distribution of enterprise factor inputs, this figure removes a special value at 573 as fitted value of enterprise factor input.

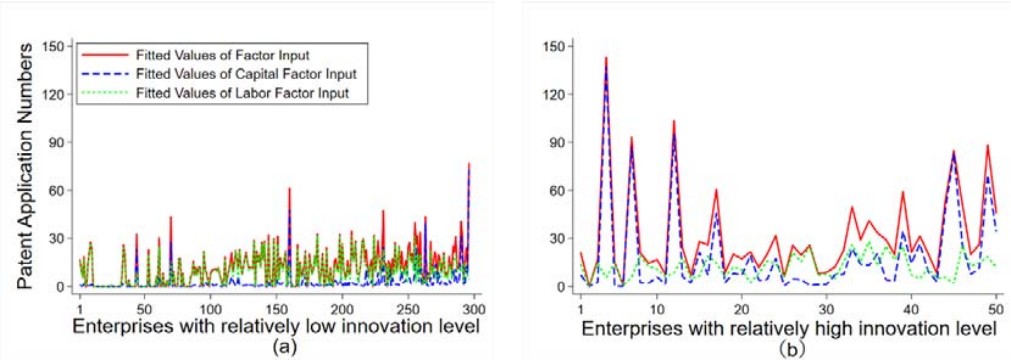

**Figure 10.** Impacts of Factor Input Indicators on Innovation Output of Enterprises with Different Innovation Levels. (**a**) Impacts of Factor Input Indicators on Innovation Output of Enterprises with relatively low Innovation Level. (**b**) Impacts of Factor Input Indicators on Innovation Output of Enterprises with relatively high Innovation Level. Note: In order to better present the distribution of enterprise factor inputs, (**a**) removes a special value at 573 as fitted value of enterprise factor input.

This indicates that enterprises with a relatively low innovation level need to improve not only their attraction to talents but also their investment in innovation R&D, in order for a strong engine of research talents for innovation. For the enterprises with higher innovation level, which are more significantly influenced by capital factor, they need to increase the capital investment in innovation on the basis of consolidating their own innovation research talent input, so as to create favorable conditions for research talents to give play to their value.

## 6. Suggestions and Future Work

Exploring the correlation between different indicators and enterprise innovation output is crucial to improve the innovation capability and output of enterprises. This paper constructs a comprehensive indicator system of enterprise innovation output influencing factors from the perspective of enterprise co-opetition, factor input and environmental factors. With a sample of 562 high-tech enterprises in Chaoyang Sub-park and Fengtai Sub-Park of Z-Park in Beijing as the research objects, this paper quantifies the key factors affecting enterprise innovation output by constructing the multiple influencing factor models. On this basis, this paper draws the following conclusions:

From the indicators of the influencing factors of enterprise innovation output, 66.7% of the indicators have a positive influence on enterprise innovation output, and 33.3% of the indicators have a negative influence. Enterprise innovation output is positively influenced by the degree of technological collaboration between enterprises, enterprises' financing capabilities, and the level of capital and labor input in innovation activities. It shows that enterprises can increase the innovation output capacity by increasing the investment of innovative talents and capital, enhancing the technical cooperation relationship with other enterprises, and improving their own financing attraction. Most of the indicators with negative effects are found in product and service competition networks and capital co-opetition networks, urging enterprises to consider the intensity of competition faced by their products and services as well as the diversity of their financing sources and investment targets in innovation activities.

It is also noticed in this study that the degree of influence of the indicators varies for different types of enterprises. Firstly, for the primary indicators, innovation outputs of enterprises with low innovation levels are mainly influenced positively by factor input and environmental factors, and negatively by network structure, while that of enterprises with high innovation levels are influenced positively by network structure and factor input. When enterprises with a relatively low innovation level improve their innovation output, they should pay attention to the optimization of their factor input and environmental factor index on the one hand and, on the other hand, they should focus on improving their own inherent innovation network structure, improve their innovation level and increase their innovation output. When improving their innovation output, enterprises with a high innovation level should increase enterprise factor input, focus on strengthening the existing enterprise innovation network structure advantages, optimize their position in the enterprise network and create conditions for the continuous improvement of the enterprise innovation level. Secondly, for the secondary indicators, under the network structure, innovation output of enterprises with a low innovation level is influenced positively by network structure indicator of the patent cooperation network (PCN), and negatively by the capital co-opetition network (CCN) and the product and service competition network (PSCN) while, for those with higher innovation level, innovation output is influenced positively by the patent cooperation network (PCN), and negatively by the product and service competition network (PSCN). The influence of patent cooperation network on the innovation output of enterprises is more important, which is more obvious in the enterprises with higher innovation level. Therefore, when optimizing the existing network structure, enterprises with a high innovation level should focus on optimizing their position in the patent cooperation network, seek more technical partners, enhance their own importance in the patent cooperation network and create good technical conditions for innovation. For enterprises with a relatively low innovation level, they should not only pay attention to the optimization of patent cooperation network structure, but also strengthen the optimization of capital competition and cooperation network structure, and product and service competition network, to reduce their negative impact on innovation output, so as to promote the innovation output of enterprises. As for the secondary indicators of factor input, both capital factors input and labor factor input exert positive influence, with the latter's influence greater for enterprises with low innovation level and the former's greater for enterprises with a higher innovation level. For the enterprises with low innovation levels, the labor factor input index has a greater positive effect on the innovation output. If enterprises with a relatively low innovation level want to improve their own innovation level, they should focus on improving their talent attraction ability, while increasing the investment in innovation research and development, so as to create a more solid foundation of scientific research talents for innovation. For enterprises with a higher innovation level, the capital factor input index has a greater positive effect on enterprise innovation. If enterprises with relatively high innovation levels want to improve their own innovation level, they should increase the investment in innovation on the basis of consolidating their

own investment in innovation and scientific research talents, so as to create conditions for better realization of the value of scientific research talents.

Built upon the above analysis, it is suggested that enterprise innovation output should be improved from the following aspects:

(1) All enterprises should pay heed to their position in the co-opetition network, and make capital of their position advantages during cooperation and competition, in exchange for the resources and information that are conducive to enterprise innovation. First, optimizing their position in the patent cooperation network can provide enterprises with the opportunity to collaborate with more enterprises for shared technology and resources and complementary advantages, so as to advance innovation output. Secondly, higher position in the capital co-opetition network gives rise to more investment and financing opportunities. Financing helps enterprises get rid of the risks they face and the impediment to innovation; making investment enables absorption and transfer of knowledge, technology and other resources, and provides the enterprise with the diverse knowledge and perspectives necessitated for innovation breakthroughs. In addition, enterprises can optimize their position in the product and service competition network (PSCN) to reduce the negative impact from market competition.

(2) There is a significant positive correlation between the innovation output of enterprises and the investment of researchers and innovation-related funds. On the one hand, increasing the capital input of enterprises can enable enterprises to obtain advanced equipment and resources, which is conducive to the innovation output of enterprises. On the other hand, it is also crucial for enterprises to develop incentive policies for researchers and improve the attractiveness of enterprises to researchers to improve the quality of innovation output. For example, for enterprises with low input level of factors and poor innovation technology, innovation not only requires capital investment, but more importantly, the introduction of talents is needed to improve the innovation ability of enterprises.

(3) Environmental factor indicators in the region or the high-tech industrial park where the enterprise is located play an important role in the development and the innovation output of the enterprise, but specific environmental factor indicators may only be applicable to a certain type of enterprise. For example, in the cases studied this paper, the environmental factor indicators of the park have much greater positive influence on the enterprises with a low innovation level than on enterprises with a high innovation level. To a certain extent, this indicates that the relevant environmental factors are more essential for the innovation development of enterprises with lower innovation levels. Therefore, governments should provide tailor-made policy support to different types of enterprises to match up with their development.

There are still some limitations in the research process of this paper, which we hope can be solved in future work. Firstly, by taking high-tech enterprises as the research object, we discussed the relationship between innovation output and its influencing factors. However, because enterprises in different industries need different resources in terms of innovation output, in order to provide more appropriate suggestions for improving innovation output capacity of enterprises, future research can study the relationship between innovation output of enterprises and its influencing factors according to different industry types. Secondly, we used the outcome indicator "number of patent applications" to measure enterprise innovation output. In later papers, scholars may consider using process indicators such as management innovation and process innovation to measure enterprise innovation output, or use the combination of outcome indicators and process indicators to measure enterprise innovation output.

**Author Contributions:** Conceptualization, L.X.; Methodology, L.S. and X.D.; Software, Y.J.; Validation, L.S., S.G., X.D. and L.X.; Formal analysis, S.G.; Investigation, L.S. and A.X.; Resources, L.S.; Data curation, S.G. and X.D.; Writing—original draft, L.S., S.G. and X.D.; Writing—review & editing, A.X.; Visualization, K.Z. and Y.J.; Supervision, L.X.; Project administration, L.X.; Funding acquisition, L.S. All authors have read and agreed to the published version of the manuscript.

**Funding:** This research was funded by the Nature Science Foundation of Shandong Province, grant number ZR2020QG056.

**Institutional Review Board Statement:** Not applicable.

**Informed Consent Statement:** Not applicable.

**Conflicts of Interest:** The authors declare no conflict of interest.

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
