# Peer review of "Influence of Enterprise’s Factor Inputs and Co-Opetition Relationships to Its Innovation Output"

_sustainability, doi:10.3390/su15010838_

Round 1
Reviewer 1 Report
The study explores a very interesting and widely read topic. Innovation is a key factor today and finding new solutions and methods is the key to growing economies. The structure of the study is appropriate and I do not recommend any changes. With regard to the literature, I would suggest that the authors try to contribute with some more literature sources to clarify the basic concepts and to establish the subject. There are a number of sources on innovation available from very diverse places, and I would suggest that these be included in the literature review. In the results section, please ask the authors to specify the source of the figures, including the item numbers. I would incorporate the appendix section into the results section of the thesis, as it shows a thorough work and helps to better understand the study.
Reviewer 2 Report
The issues raised by the Authors are topical and may be interesting for the reader. It can be a source of inspiration for both, theoreticians and practitioners. It can be a starting point for further scientific research.
The text is comprehensive and clear, nevertheless, it requires corrections.
The title: Uderstandable, but perhaps it should not necessarily be phrased as a question. I suggest changing it to a sentence equivalent.
Abstract: The abstract should be completed with the purpose, the other elements are included. Other than that, the summary as a whole is written correctly and concisely.
Keywords: Ccorrect.
The structure of the text lacks a clearly distinguishable methodology part. The structure of the text should be improved.
The introduction is correct and interesting. Authors put the topic in context, but some information is, in my opinion unnecessary (the part about Nokia, Microsoft and Apple). The purpose of the article should also be added in this section.
The literature review was carried out correctly, the Authors reviewed the literature in terms of the analyzed aspects - the contribution of factors, competitive and cooperative relations between enterprises and the market environment in a coherent and logical manner. The literature is relevant to the researched subject and up-to-date, most of being from the last 5 years. The research gap has been sufficiently indicated.
Points 3 and 4 should be formed as a larger whole - the methodology. In point 3.1 (research object) lacks the characteristics of the research sample, it is not indicated when the research was carried out.
Section 5 (called “results and discussions”) describes the conclusions in a coherent and clear manner, but does not include a discussion. I suggest renaming it to “conclusions” or having a discussion of the results obtained.
Point 6 (Conclusions and suggestions) should also include limitations and directions for further research. This section should be completed.
Reviewer 3 Report
Dear authors,
I am honored to read such a meaningful article. At the same time, I also found some problems. Some questions are as follows:
1. In the 34th citation on page 5 of section 2.3, the original document is missing.
2. There are two 3.2.2 sections on page 6. Should the second one be "3.2.3 Co-opetition Relationship between Enterprises"?
3. On page 11 of section 5.1, that is, the analysis of Table 2, you have explained only some of the factors. Do you need to explain all of them, or explain why only these factors are explained? What are the representativeness of these factors?
4. In line 548 on page 13 of section 5.2, how did you get the 1.6 times and 6 times mentioned and as mentioned in the later analysis? Are they calculated? I suggest you explain them in detail.
5. In line 572 on page 13 of section 5.3.1, should "PSC" be changed to "PCN"?
6. In Figure 5 (a) on page 14 of section 5.3.1, "CCCN" should be changed to "CCN".
7. The conclusions and suggestions should be more substantial, especially the suggestions.
Best regards,
Reviewer
Reviewer 4 Report
Please see attachment
